# Impact of Geraniol and Geraniol Nanoemulsions on *Botrytis cinerea* and Effect of Geraniol on Cucumber Plants’ Metabolic Profile Analyzed by LC-QTOF-MS

**DOI:** 10.3390/plants11192513

**Published:** 2022-09-26

**Authors:** Nathalie N. Kamou, Natasa P. Kalogiouri, Panagiota Tryfon, Anastasia Papadopoulou, Katerina Karamanoli, Catherine Dendrinou-Samara, Urania Menkissoglu-Spiroudi

**Affiliations:** 1Pesticide Science Laboratory, Faculty of Agriculture Forestry and Natural Environment, School of Agriculture, Aristotle University of Thessaloniki, 54124 Thessaloniki, Greece; 2Laboratory of Analytical Chemistry, Department of Chemistry, Aristotle University of Thessaloniki, 54124 Thessaloniki, Greece; 3Laboratory of Inorganic Chemistry, Department of Chemistry, Aristotle University of Thessaloniki, 54124 Thessaloniki, Greece; 4Laboratory of Agricultural Chemistry, Faculty of Agriculture, School of Agriculture, Forestry and Natural Environment, Aristotle University of Thessaloniki, 54124 Thessaloniki, Greece

**Keywords:** geraniol-loaded nano emulsions, LC-QTOF-MS, bioactive substances, antifungal activity

## Abstract

In the present study, the bioactive substance geraniol was tested in vitro and in planta against *B. cinerea* on cucumber plants, and the changes in the metabolic profile of cucumber plants inoculated with the pathogen and/or treated with geraniol were monitored by a novel LC-QTOF-MS method employing target and suspect screening. The aforementioned treatments were also studied for their impact on membrane lipid peroxidation calculated as malondialdehyde (MDA) content. Additionally, geraniol-loaded nanoemulsions (GNEs) were synthesized and tested against *B. cinerea* as an integrated formulation mode of geraniol application. The EC_50_ values calculated for geraniol and GNEs against *B. cinerea* were calculated at 235 μg/mL and 105 μg/mL, respectively. The in planta experiment on cucumber plants demonstrated the ability of geraniol and GNEs to significantly inhibit *B. cinerea* under greenhouse conditions. The LC-QTOF-MS analysis of the metabolic profile of the cucumber plants treated with geraniol demonstrated an increase in the concentration levels of myricetin, chlorogenic acid, and kaempferol rhamnoside, as compared to control plants and the presence of *B. cinerea* caused an increase in sinapic acid and genistein. These compounds are part of important biosynthetic pathways mostly related to responses against a pathogen attack.

## 1. Introduction

Cucumber (*Cucumis sativus* L.) is widely cultivated throughout the world, with the total production being equal to 87.805.086 tons worldwide in 2019 [1]. Cucumber cultivation is considered a demanding procedure regarding agrochemicals since the plant has high requirements for fertilizers and it can be affected by numerous pests and pathogens during its growth [2]. 

*Botrytis cinerea* (teleomorph *Botryotinia fuckeliana*), the causal agent of gray mold, is responsible for the quality and quantity reduction of economically important vegetables such as cucumber. The pathogen affects all the aboveground parts of the host, destroying the leaves, the flowers, the stem, and the fruits. Recently, *B. cinerea* was listed as the second most important pathogen worldwide, based on the economic importance and the scientific interest it causes [3]. The current agricultural practices are based on the use of mainly conventional agrochemicals, although the interest for effective alternative ecofriendly control means is increasing, especially in economically important crops in the frame of IPM strategies [4]. Fungicide application against *B. cinerea* is the prevailing choice of control, considerably increasing the cost of production. More specifically, it is estimated that the total cost to control *B. cinerea,* with all available means using an integrated approach, can exceed EUR 1 billion/year [3]. In addition to the cost, fungicide resistance is another major drawback of the use of conventional synthetic agrochemicals since they have become ineffective [5,6]. The vicious cycle of fungicide resistance urges the producers to increase the dosage of the agrochemicals in many cases, thus aggravating the environmental pollution with chemical residues. Since *B. cinerea* is considered a high-risk pathogen regarding fungicide resistance [5], developing an alternative control method that avoids the use of conventional synthetic fungicides is a necessity. 

The use of combinations of different essential oils and their isolated components as fungicides has gained the interest of the research community during the last decades. This attention is attributed to their antifungal, antibacterial, antiviral, insecticidal, and antioxidant properties [7]. Terpenoids are components of plant essential oils with important antimicrobial properties [8]. Geraniol is a monoterpene alcohol and the major component (47.08%) of lemon thyme (*Thymus citriodorus*) essential oil [9], showing significant antimicrobial activity. The multiple important biological activities of geraniol (insecticidal, acaricidal, bactericidal, fungicidal, antioxidant, and anti-inflammatory), combined with its low toxicity against mammals, make this monoterpene suitable for further research as an alternative to conventional fungicides [10]. Chen et al. [10] reported that the antibacterial properties of monoterpene alcohols, such as linalool, nerol, citronellol, and geraniol, are more adequately documented as compared to their antifungal activity, especially regarding human pathogens. However, the hydrophobic and volatile nature of geraniol underlines the necessity to find a proper type of formulation due to its low solubility in aqueous means. The nanoencapsulation process of essential oils through the preparation of aqueous nano-dispersions is the easiest way to formulate, handle, and of low cost to overcome these problems. Thus, the application of nanoencapsulated geraniol can help alleviate this drawback since it has been proved that nanoparticles (NPs) allow gradual release of the hydrophobic compounds into the environment and also could improve their solubility [11]. Yegin et al. [11] demonstrated that geraniol-loaded polymeric nanoparticles ranging in size from 26 to 412 nm successfully inhibited two important bacterial human pathogens and the NPs maintained the antimicrobial activity for 24 h since this was the time needed for the sustained release of geraniol. Moreover, stability and efficacy are furtherly supported when these highly volatile compounds are applied as an oil-in-water nanoemulsion [12]. 

In order to be used in agricultural practice, natural plant compounds must be biosafe. Several studies focus on the use of geraniol as a drug, a cosmetic, and a detergent product [10,13], and it is also included in EU Pesticides Database, but there are limited studies investigating its use against plant pathogens and its further effect on plants. Geraniol has been studied for its cytotoxic and genotoxic effects when applied on root tip meristem cells from *Hordeum vulgare* at a high concentration (500 μg/mL) [14]. Moreover, Ložiene and Vaiciulyte [15] demonstrated the phytotoxic aspect of geraniol on *Hypericum perforatum* and *Phleum pratense,* providing further information on the effect that geraniol has on plants. In this context, it is interesting to investigate the effect of geraniol on the plants’ metabolic profile since chemical control means definitely affect the plant metabolism [16,17]. 

Liquid Chromatography Quadrupole Time-of-Flight Mass Spectrometry (LC-QTOF-MS) is frequently used in order to identify metabolic compounds of plants belonging to the *Cucurbitaceae* family [18,19,20,21,22]. The high resolving power of QTOF-MS allows the simultaneous analysis of a wide range of naturally occurring constituents through target and non-target screening strategies [23].

The aim of this study was to investigate the antifungal activity of geraniol against *B. cinerea* both in vitro and in planta on cucumber plants and to study the impact of the pathogen, geraniol, and their combination on the metabolic profile of cucumber plants. The metabolome of the cucumber plants was analyzed by a novel LC-QTOF-MS method, employing target and suspect screening. Moreover, the impact of these treatments was further studied by estimating membrane lipid peroxidation calculated as malondialdehyde (MDA) content. These results provide insights into the biochemical and metabolic processes deployed by the plant when triggered by the aforementioned treatments. Furthermore, geraniol—loaded Nano Emulsions (GNEs from hereafter) were synthesized, characterized, and evaluated against *B. cinerea* as an integrated mode of geraniol application that ensures the translocation, gradual release, and stability of this compound.

## 2. Results

### 2.1. GNEs: Characterization and Pharmacokinetic Study 

The calculated percentage of encapsulation efficiency (EE, %) for entrapped geraniol in GNEs was found to be 57%, while the loading capacity (LC, %) was found to be 14%, indicating the percent of geraniol entrapped within the sodium dodecyl sulfate (SDS) during nanoemulsion synthesis. In order to apply the emulsion system, it is necessary to maintain particle size throughout its designated shelf life. Therefore, the stability of GNEs was measured by examining the change in the particle size during the storage period (0 h and 96 h). Figure 1 depicts the size distribution for GNEs with an average diameter of 256 nm and 296 nm at 0 h and 96 h, respectively, revealing a stable emulsion formation on the nanoscale and a good homogeneity (PDI = 0.35). The ζ-potential was measured at −58 mV and −61 mV at 0 h and 96 h, respectively, showing a negatively charged surface. 

The release of entrapped geraniol from GNEs and the release profile of native geraniol were estimated via the dialysis bag approach. A pharmacokinetic study was determined to analyze the release mechanism through zero order, first order, Higuchi, and Korsmeyer–Peppas models and their linearity (Figure 2). Figure 2A presents the zero-order model regarding a profile of native geraniol released over 55% in 4 h, while the same percentage was reached for GNEs after 72 h verifying the sustained release. The fitting values were R^2^ = 0.9758 and R^2^ = 0.8889, respectively. Overall, the initial release of geraniol from the GNEs was slow but gradual throughout 0 to 24 h. No burst release effect was observed in the active ingredient profile in contrast with that of native geraniol. The SDS micelle matrix acts as a barrier to control the release of geraniol and the drug’s physicochemical properties. Figure 2B illustrates the dependent first-order kinetics. The fitting values of native geraniol and GNEs were provided at R^2^ = 0.9116 and R^2^ = 0.9296, verifying a concentration-dependent release profile. The release plateau at 14 h follows the first order of kinetic. The Higuchi model (Figure 2C) with R^2^ = 0.9243 and K_H_ = 27.36 and R^2^ = 0.9067 and K_H_ = 12.26 values, respectively, confirmed the diffusion mechanism of the release. Furthermore, the Korsmeyer–Peppas model was determined (Figure 2D) to categorize the diffusion mechanism based on the slope (N value). The N value of native geraniol (N = 0.52) and GNEs (N = 0.46) was estimated, revealing a non-Fickian anomalous diffusion release of geraniol from the native form and a Fickian diffusion release of geraniol from GNEs. 

### 2.2. In vitro Antifungal Activity

The antifungal activity of geraniol and GNEs against *B. cinerea* was initially investigated in vitro, and EC_50_ values were estimated (Table 1) based on the dose–response curves (Figure 3 and Figure 4). The bioassays clearly demonstrate that both substances successfully inhibit *B. cinerea* growth in a dose–response manner. The mean EC_50_ values calculated for geraniol and GNEs against *B. cinerea* were 235 μg/mL and 105 μg/mL, respectively.

### 2.3. In Planta Antifungal Activity

Disease severity was estimated on cucumber plants in order to demonstrate the antifungal activity of geraniol and GNEs, as described above. Results indicated that both treatments caused a significant decrease in disease index as compared to the *B. cinerea* control treatment but did not differ significantly compared to the chemical control, as shown in Figure 5 and Figure 6. No phytotoxic effects were observed on cucumber plants sprayed either with geraniol, GNEs, or chemical control (Luna SC). 

### 2.4. Biochemical Assays

MDA is used as an important indicator of membrane lipid peroxidation in plants. In this study, *B. cinerea* treatment caused an increase of 182% compared to control plants (Figure 7). By contrast, MDA content was not altered in the plants treated with geraniol after inoculation with *B. cinerea,* with the concentration remaining at control levels (Figure 7). Interestingly though, plants treated with geraniol alone also showed significantly increased levels of MDA (88.8%).

### 2.5. LC-QTOF/MS Analysis

The identification results regarding the target compounds are presented in Table 2, and the identified suspect metabolites are listed in Table 3. The EICs, MS, and MS/MS spectra are presented in Appendix A. The MS/MS spectrum of each analyte was compared with online libraries, such as MassBank [24] and FooDB [25], and literature records, as well. All the tentative identification criteria are summarized in Table 3. After the identification, the suspect compounds were semi-quantified on the basis of the target compounds belonging to the same chemical class [26]. Six of the compounds that were identified showed a significant increase after treatments with geraniol, inoculation with *B. cinerea,* and their combination, as compared to control plants. In target screening, sinapic acid was determined in cucumbers at concentrations ranging from 0.5876 (Cucumber control) to 0.9584 mg/kg (*B. cinerea* + geraniol), and myricetin at concentrations ranging from 0.2156 (Cucumber control) to 0.2178 mg/kg (Geraniol control) (mean of five replicates Table 4). Regarding the substances that derived from the suspect screening, chlorogenic acid, which was semi quantified using the calibration curve of caffeic acid, was identified at concentrations ranging from 100.59 (Cucumber control) to 152.54 mg/kg (Geraniol control). Genistein that was semi-quantified using the calibration curve of rutin was determined in cucumbers at concentrations ranging from 0.7252 (Cucumber control) to 0.7359 mg/kg (*B. cinerea* + geraniol) (Table 4). 

Kaempferol rhamnoside was semi-quantified using the calibration curve of rutin. It was determined at concentrations from 0.9080 (Cucumber control) to 1.7620 (*B. cinerea* + geraniol) (Table 4). 

## 3. Discussion

The use of natural bioactive compounds with low phytotoxicity against important plant pathogens such as *B. cinerea* could be an alternative practice to the current overuse of synthetic fungicides. In the present study, we demonstrated the ability of both native geraniol and geraniol-loaded Nano Emulsions (GNEs) to significantly reduce the disease index caused by *B. cinerea* on cucumber plants. The effect of essential oils as natural and promising alternatives to synthetic fungicides is already known. More specifically, the use of essential oils from *Cymbopogon martinii,* which mainly consisted of geraniol (83.82%), geranyl acetate (7.49%), linalool (2.48%), and caryophyllene (1.33%), demonstrated very promising results against *B. cinerea* post-harvest infections of strawberries [31]. An 83% reduction in disease incidence occurred when *C. martinii* essential oils were applied as vapor at a 10% concentration, indicating the possibility of using them as an alternative to post-harvest fungicides against *B. cinerea* [31]. Recent research has focused on finding natural antimicrobial compounds in order to inhibit *B. cinerea*, the causal agent of serious and economically damaging pre- and post-harvest losses [31,32,33,34]. The interest is not limited only to the essential oils since their individual constituents have also been investigated as potential antifungal agents against *B. cinerea* [35,36,37,38]. As regards the mode of action of lipophilic volatile mixtures and their constituents, such as geraniol, it has been found that it is mostly attributed to their ability to disturb cell membrane integrity and cause metabolic changes [39,40]. In this context, geraniol could display a double inhibitory role since its antifungal activity targets important human and plants’ pathogens through different modes of action, such as increase in ROS accumulation in *A. flavus* or impairment of cell membrane in *A. ochraceus* [40]. As demonstrated in the in vitro bioassays, the GNEs were proven to be more effective against the pathogen compared to the compound itself. This result can be attributed to the fact that nanoemulsions provide a controlled release of geraniol that increases the efficiency and availability of the volatile compound. In the present study, the significant inhibition of *B. cinerea* by the GNEs in the planta bioassay was achieved at a concentration of 210 μg/mL (EC_50_ 105 μg/mL). GNEs presented satisfactory PDI, EE%, and LC% values based on the aforementioned results, making them suitable for further in vitro and in planta study as a stable nano-delivery system. The relative slight increment in the mean particle diameter of the sample during the storage period can be attributed to the coalescence of the droplets due to Ostwald ripening since geraniol has some degree of water solubility because of the hydroxyl group in their phenolic structure [41]. Maintenance of particle size and avoidance of the destabilization phenomena attributed to Ostwald ripening could ensure longer shelf life and a more effective application of nanoemulsions when needed [42]. Low PDI values relate to monodispersity and have been indicated previously when SDS was used as an emulsifier for geraniol (PDI = 0.277) [12]. However, higher PDI values were recorded when mixed fractions of the oil phase (geraniol: carvacrol) were used [12]. Small-molecule emulsifiers such as SDS and Tween could be more effective in making tiny/nano droplets than biopolymers such as caseinate and β-lactoglobulin [43]. The SDS:geraniol ratio is also one of the factors that can affect the EE% and LC%. Geraniol was successfully entrapped in the formulation of SDS nanoemulsions, while SDS is an anionic surfactant that penetrates the oil droplet and forms stable core-shell nanoparticles, thus presenting an effective biocompatible coating [44]. At high concentrations of SDS, as in the present study, the charge density on the droplet is increased, resulting in an increase in migration times for hydrophobic solutes (like geraniol). Specifically, the negatively charged heads of SDS to oil (geraniol) droplets’ surface increased the electrostatic repulsion between droplets, forming a homogeneous and stable nanoemulsion formulation [45,46]. The results of the kinetic study showed that nanoemulsions using SDS as an emulsifier could be studied as a geraniol-delivery system due to their good long-term stability of geraniol. Furthermore, no phytotoxicity was detected in cucumber plants, which is rather rare and promising since plant toxicity phenomena caused by the application of nanoparticles on cucumber were reported in the past [47]. To the best of our knowledge, this is the first research project focused on the in vitro and in planta effect of geraniol and GNEs against *B. cinerea*. 

According to the metabolic profiling, the cucumber plants treated with geraniol demonstrated an increase in myricetin, chlorogenic acid, and kaempferol rhamnoside as compared to control plants. Since phenolic acids and flavonoids are compounds that can be effective as phytoalexins and/or phytoanticipins [48] in response to pathogen attack, their presence, even in the absence of the pathogen, indicates plant alertness towards pathogenicity induced by geraniol. Detection of these phenolics in the leaves of cucumber after fungus infection and treatment with geraniol in similar (myricetin, chlorogenic acid) or even in greater quantity (sinapic acid and kaempferol rhamnoside) indicates their crucial role in plant chemical response activation. Similarly, accumulation of chlorogenic acid was recorded in maize plants infected with *Colletotrichum graminicola* [49]. It is worth mentioning that transgenic tomato plants that are able to overproduce chlorogenic acid demonstrated resistance toward the bacterial pathogen *Pseudomonas syringae* [50]. Moreover, chlorogenic acid displays significant antifungal activity and effectively inhibits the growth of necrotrophic pathogens such as *B. cinerea* [51]. Similarly, kaempferol rhamnoside is considered a resistance-related metabolite against *Fusarium graminearum* [52]. Thus, the antifungal activity of the detected plant phenolic compounds, together with the activity of geraniol itself, could possibly explain the inhibition of *B. cinerea* and the reduction of disease index in the infected plants treated with geraniol.

Interestingly, in cucumber plants inoculated with the pathogen, an increase in sinapic acid and genistein was found as a response to *B. cinerea* presence. These compounds are part of important biosynthetic pathways associated with plant defense mechanisms. In particular, genistein is one of the main phytoalexins accumulated in soybean plants after a pathogen attack [53] and serves as a common precursor in the biosynthesis of antimicrobial phytoalexins and phytoanticipins, and sinapic acid biosynthesis leads to accumulation of lignin, which rigidifies and strengthens the cell wall structure after pathogen infection [54,55,56]. 

At this point, it is worth mentioning that MDA, a useful indicator of oxidative stress, exhibited its higher value in plants infected with the fungus. Moreover, a slight increase in this molecule compared to control was also reported after geraniol application, while its content did not differ from control in the case of infected plants treated with geraniol. This last finding is probably connected with the reduced cell damage of leaves, as also indicated by the low disease index in the same treatment. On the contrary, increased MDA levels upon fungus application are probably connected with increased oxidative stress, which is further supported by the enhanced disease index detected in this treatment. The slightly higher MDA content in uninfected plants after geraniol application points out its possible role as a signal molecule that can increase plant alertness for amplified defense responses upon stress initiation [57]. It is well documented lately that increased levels of MDA may represent acclimation processes rather than damage since MDA can exert a positive role by activating regulatory genes involved in plant defense and development [58] and by acting also as a signaling molecule and regulator of essential biological functions [59,60]. The obtained results corroborate others [61,62], according to which clove essential oil effectively controlled the disease incidence of blue mold decay in citrus fruit by suppressing MDA accumulation, and coriander oil significant decreased the MDA content of infected flax leaves by Powdery Mildew.

To summarize, secondary metabolites basically act as regulatory and signaling molecules affecting several plant physiological responses. The plant’s priming mechanisms closely depend on the type of biotic and abiotic stress it is exposed to and is a dynamic response that combines several mechanisms and biosynthetic pathways. The use of bioactive substances, such as geraniol, that induce plant responses could be an important step towards decreasing the use of conventional pesticides. Due to the limited number of studies regarding the effect of geraniol on plant priming and its use as an alternative control mean for important diseases, it is important to investigate the best strategy to increase productivity and protect human health and the environment at the same time. The use of bioactive substances in combination with nanotechnology in crops could see to that end. Of course, the potentially toxic effects of nanopesticides on the environment and on consumers’ health should be addressed in order to propose a holistic and safe approach to crop production. 

## 4. Materials and Methods

### 4.1. Preparation of GNEs

All the reagents were of analytical grade and were used without any further purification: Sodium Dodecyl Sulfate (CH_3_(CH_2_)_11_SO_4_Na), SDS (Sigma-Aldrich, ≥99%, *M* = 288.38 g/mol), geraniol, trans-3,7-Dimethyl-2,6-octadien-1-ol (Sigma-Aldrich, 98%, *M* = 154.25 g/mol), and diethyl ether (Merck, ≥99%, *M* = 74.12 g/mol). 

Geraniol Nanoemulsions (GNEs) were prepared via a simple nanoemulsion-based approach reported previously [63], with slight modifications. Briefly, pure commercial geraniol (111 μL) was dissolved in diethyl ether (3 mL), followed by the addition of sodium dodecyl sulfate (SDS) (19.5 mM) and ddH_2_O (30 mL). The mixture was kept in a closed vial and emulsified by sonication treatment for two hours at ambient temperature (<30 °C). Subsequently, the vial was opened, and diethyl ether was evaporated to be slowly removed. The prepared oil-in-water (O/W) nanoemulsions were stored in room temperature conditions (25 °C) for further characterization. 

#### 4.1.1. Physicochemical Characterization of Geraniol Nanoemulsions

The mean diameter (d.nm), polydispersity index (PDI), and zeta-potential (ζ-potential) of the GNEs were determined at 25 °C by dynamic light scattering (DLS) on a Zetasizer Nano ZS (Malvern Panalytical, UK). Nanoemulsions were diluted with distilled water, and the analysis was conducted in triplicate at 0 and 96 h to verify their stability. UV-Visible measurements were carried out with a double beam UV-Visible spectrophotometer U-2001 Hitachi. The percentage of encapsulation efficiency (EE, %) and the loading capacity (LC, %) were calculated by the following equations [64,65]: $\mathrm{EE}\;\left(\%\right)=\frac{\mathrm{total}\;\mathrm{amount}\;\mathrm{of}\;\mathrm{geraniol}\;-\;\mathrm{amount}\;\mathrm{of}\;\mathrm{free}\;\mathrm{geraniol}}{\mathrm{total}\;\mathrm{amount}\;\mathrm{of}\;\mathrm{geraniol}}\times 100\%$ and $\mathrm{LC}\;\left(\%\right)=\frac{\mathrm{mass}\;\mathrm{of}\;\mathrm{loaded}\;\mathrm{geraniol}\;}{\mathrm{mass}\;\mathrm{of}\;\mathrm{sample}}\times 100\%$

#### 4.1.2. Geraniol Release

For the drug release study, 400 μL of the GNEs were placed into a Pur-A-Lyzer Midi 1000 dialysis membrane. The membrane floated in 40 mL of phosphate buffer solution (PBS; pH = 7.2) and was shaken in a stirrer (200 rpm) over 96 h. Aliquots (2 mL) of the medium were periodically removed (0, 1, 2, 3, 4, 8, 16, 24, 48, 72, and 96 h) and replaced with fresh PBS solution. The same procedure was followed for native geraniol, but in ethanol solution, since in this solvent, it was dispersed to be applied in bio-assays. The release data were evaluated using the mathematical models: Zero order, First order, Higuchi, and Korsmeyer–Peppas [66,67]. Three independent replications were carried out for each measurement. 

### 4.2. In Vitro Antifungal Activity

*B. cinerea* strain B05 was obtained from the culture collection of the Laboratory of Plant Pathology, School of Agriculture, Faculty of Agriculture, Forestry and Natural Environment, Aristotle University of Thessaloniki, and routinely kept on potato dextrose agar (PDA, BD Difco) plates at 25 °C. Geraniol analytical standard (985) was purchased from Sigma-Aldrich (CAS Number: 106-24-1). A stock solution of geraniol was prepared by dilution in 70% ethanol 99.8% purchased from Sigma-Aldrich (CAS Number: 64-17-5) at 1:3 ratio and used for the preparation of working solutions.

Antifungal activity was assessed with the incorporation of geraniol and GNEs in the PDA medium on which the pathogen was added, and its growth was measured daily. Both geraniol and the GNEs were dispersed in the medium when the temperature was around 40 °C to avoid degradation due to high temperature. Proper amounts of geraniol stock solution were added to PDA in order to obtain seven (7) concentrations (150, 175, 200, 225, 250, 300, and 350 μg/mL) and investigate the dose–response inhibition of the pathogen. Calculated amounts of the GNEs—based on the loading capacity of geraniol—were added to the PDA medium to obtain 7 concentrations of 25, 75, 100, 125, 150, 250, and 500 μg/mL. The mixtures of each concentration of both geraniol and GNEs were transferred in an ultrasonic bath (Transsonic 460 h, Elma GmbH & Co KG, Singen, Germany) for 60 s to ensure better dispersal. *B. cinerea* was inoculated as 0.5 cm agar plugs on the solidified medium and then incubated at 25 °C until full growth of the control treatment. In addition, coating surfactant of the GNCs (Sodium Dodecyl Sulfate—SDS) and ethanol were also tested separately at the highest concentration in order to exclude any possible inhibition of the fungal growth attributed to these materials. Each treatment had five technical replications, and the experiment was repeated twice. 

The growth diameter of *B. cinerea* was measured when the control treatment reached full growth, and the growth inhibition rate was calculated using the equation $\mathrm{Inhibition}\;\left(\%\right)=\frac{\mathrm{Diameter}\;\mathrm{of}\;\mathrm{control}-\mathrm{diameter}\;\mathrm{of}\;\mathrm{treatment}}{\mathrm{Diameter}\;\mathrm{of}\;\mathrm{control}}\times 100\%$ [67]. The ability of geraniol and GNEs to inhibit fungal growth was estimated based on mean EC_50_ values (half-maximal effective concentration causing 50% inhibition of mycelial growth) using graded dose–response curves.

### 4.3. In Planta Antifungal Activity

Cucumber plants cv. Bamboo in the second leaf stage were used to investigate the ability of geraniol and GNEs to inhibit *B. cinerea* B05. Plants were transplanted in 100 mL pots, irrigated regularly, and kept under greenhouse conditions at 20–25 °C with a 16/8 h photoperiod cycle and 60–70% RH, without any pesticide or fertilizer application. Eight (8) plants were used for each treatment, and the experiment was repeated thrice. 

The pathogen was inoculated according to Elad et al. [68] with slight modifications using droplets (10 μL) of conidial suspension (2 × 10^6^ sp/mL) on the upper leaf part, and the application solution consisted of water: PDB (Potato Dextrose Broth) at 1:1 ratio and 0.1% Tween 20 as a surfactant. 

Regarding the treatments of geraniol and GNEs, plants were spayed at concentrations causing 100% inhibition of *B. cinerea*, calculated based on the EC_50_ values (470 and 210 μg/mL, respectively). The geraniol spraying solution was prepared by properly diluting the stock solution in water containing 0.1% DMSO and 0.1% Tween 20. Symptom evaluation was conducted daily for 96 h. GNEs were added as an oil-in-water solution.

Non-treated plants and plants treated with Luna Experience SC, at the highest recommended dose, were used as a control and a chemical control, respectively. In addition, plants sprayed with sole SDS at the same concentration used for the GNEs synthesis were tested for phytotoxicity.

The extent of the disease caused by *B. cinerea* on the leaves was evaluated according to a disease index (DI) based on symptom development as follows: 1: no symptom; 2: 1–12% of droplet area necrotic; 3: 13–25% necrosis; 4: 26–50% necrosis; 5: 51–100% necrosis; 6: necrotic area exceeding droplet diameter by up to 1 mm; 7: necrosis exceeding droplet diameter by 1–3 mm; 8: necrosis exceeding droplet diameter by more than 3 mm [68] (Figure 8).

### 4.4. Malondialdehyde (MDA) Content

Membrane lipid peroxidation as malondialdehyde (MDA) content was estimated by the thiobarbituric acid (TBA) method, as described previously [69]. In brief, cucumber leaf tissue powder (200 mg) was homogenized in 600 μL of 0.1% trichloroacetic acid (TCA), followed by centrifugation at 15.000 rpm for 20 min at 4 °C. Then, 0.5 mL of supernatant was mixed with 1.5 of 0.5% TBA in 20% TCA. The mixture was incubated at 95 °C for 25 min in a water bath, and the reaction was terminated in an ice bath. The reaction mixture was then recentrifuged for 15.000 rpm for 5 min at 4 °C, and the absorbance of the supernatant was measured at 532 nm and corrected at 600 nm. The concentration of MDA was determined using its extinction coefficient of 155 μM^−1^ cm^−1^. Results were expressed as micromole of MDA per gram of fresh weight.

### 4.5. LC-QTOF-MS Analysis

#### 4.5.1. Standards and Reagents

Methanol (MeOH), water (H_2_O) LC-MS grade, and formic acid 98–100% were purchased from Merck (Darmstadt, Germany). For the determination of phenolic compounds, p-coumaric acid 98%, caffeic acid 98%, sinapic acid 98%, ferulic acid 98%, apigenin 98%, luteolin 98%, myricitrin 98%, and myricetin 98% were purchased by Sigma-Aldrich (Stenheim, Germany). Stock standard solutions of all the standard compounds were prepared in methanol at 1000 μg/mL and stored at −20 °C. Working solutions were prepared by dilution of the stock solutions in methanol. Mixtures of 30 mg/L containing all the standard analytes were prepared by diluting the stock standard solutions in LC-MS grade MeOH.

#### 4.5.2. Sample Preparation

In order to investigate which metabolites are affected by the pathogen, geraniol, and/or their combination, 5 samples of 200 mg of cucumber leaf tissue from each treatment (cucumber control, *B. cinerea* control, geraniol control, and *B. cinerea* + geraniol) were immediately homogenized in liquid nitrogen and stored at −80 °C for metabolite analysis. Frozen samples were transferred in 2-mL screw cap tubes, and 1 mL of MeOH:water (80:20, *v*/*v*) was added to extract the bioactive compounds from the plant matrix [70]. Then, the mixture was vortexed for 1 min. The mixtures were added in an ultrasonic bath (Transsonic 460 h, Elma GmbH & Co KG, Singen (Hohentwiel), Germany) for 10 min, and centrifugation followed at 8000 rpm at 4 °C within 5 min. After transportation in a new tube, each sample was recentrifuged and then filtered using a nylon syringe filter with a pore size of 0.22 μm.

#### 4.5.3. Chromatographic Analysis

An ExionAC LC system (SCIEX, MA), equipped with two pumps, solvent degasser, autosampler, and controller, was used for the analysis of the metabolites. The LC system was interfaced to an X500R Q-TOF mass spectrometer (SCIEX, Framingham, MA) equipped with electrospray ionization (ESI) turboVTM source operated in the negative ion mode. A data-dependent acquisition (IDA) electrospray ionization mode was used to acquire TOF-MS and TOF-MS/MS data. The chromatographic separation was carried out in a Fortis C18 column (100 mm length, 2.1 mm i.d, 2.6 µm particle size) purchased from Fortis (Cheshire, United Kingdom) equipped with a pre-column of Fortis SpeedCore C18 (10 × 2 mm, 2.6 μm, Fortis, Cheshire, United Kingdom), thermostated at 40 °C. The gradient system consisted of: (A) 90% H_2_O, 10% MeOH with 0.1% formic acid, (B) 100% MeOH with 0.1% formic acid. The elution gradient program started with 1% of organic phase B (flow rate 0.2 mL min^−1^) for one min, gradually increasing to 39% for the next 4 min, and then increasing to 95% (12–15 min) and remaining constant for the following 3 min (flow rate 0.4 mL min^−1^). Then, the organic phase increased gradually to 99% at a flow rate of 0.2 mL min^−1^ within 1 min and remained constant for another 5 min (16–21 min). The initial conditions (1% B–99% A) were restored within 0.1 min (flow rate decreased to 0.2 mL min^−1^) to re-equilibrate the column for 5 min prior to the next injection.

The ESI interface operated in a negative mode with the following settings: spray voltage of-4500 V, 550 °C heater gas temperature, and 80 V declustering potential. The collision energy of 45 V and a collision energy spread of 15 V were set to acquire MS/MS spectra. A cluster solution provided by SCIEX was used for external validation. Additionally, the calibration solution was injected at the beginning of each run for internal calibration and once per five samples during batch acquisition. Mass spectra were recorded in the range from 50 to 1000 Da at an accumulation time of 0.25 s. MS/MS experiments were conducted using IDA-dependent mode at an accumulation time of 0.08 s for the 10 most abundant precursor ions per full scan. The SCIEX OS software was used for sample acquisition monitoring. Extraction ion chromatograms (EICs) were created using the SCIEX OS software and setting the following parameters: mass accuracy window of 5 ppm; signal to noise threshold of 3; minimum area threshold of 1000; minimum intensity threshold of 500. The identification of the analytes was based on the accurate mass, isotopic pattern, and the comparison of the retention time and MS/MS spectra of the target analyte with reference standards. For the analytes that there were no reference standards available, in suspect screening, tentative identification was performed based on the accurate mass, isotopic pattern, and MS/MS fragments.

#### 4.5.4. Target and Suspect Screening Strategies

Target and suspect screening workflows were applied for the determination of metabolites in the samples, according to Kritikou et al. [70]. In target screening, the identification was carried out using standard solutions for confirmation. For the quantification of the results, calibration curves were constructed and found linear over the range of 0.01–5 mg/kg (Table 5). In suspect screening, a suspect list was generated from the literature [19,20,21,22,23,71] and references therein, including all the antioxidant compounds that have already been identified in other plants belonging to the *Cucurbitaceae* family, in order to scan their presence in the analyzed samples. For the identification of the analytes, the exact mass, isotopes, and adducts were calculated, while the MS/MS fragments were examined and compared with online libraries, such as MassBank [24] and FooDB [25], and literature records, as well. After the identification, the suspect compounds were semi-quantified on the basis of the target compounds of the same class having similar structures [23].

### 4.6. Statistical Analysis

Regarding the in vitro bioassays, EC_50_ values of geraniol and GNEs against *B. cinerea* were calculated using a nonlinear dose–response curve and using ten replicates per GNEs concentration (5 per concentration and repeated twice) using Origin Pro 8 (Data Analysis and Graphing Software). The in planta experiments were analyzed by analysis of variance (ANOVA), based on the completely randomized design (CRD), and mean values were computed from the respective replicates. Statistical analysis, a one-way analysis of variance followed by Tukey’s post hoc test (*p* ≤ 0.05), was conducted using SPSS v 25.0 software (SPSS, Chicago, IL, United States). The statistical analyses for physicochemical characterization were performed through Origin Pro 8 (Data Analysis and Graphing Software) and SPSS v 25.0 software (SPSS Inc. Chicago, IL, USA). 

## 5. Conclusions

This is the first attempt to study the effect of geraniol and GNEs against *B. cinerea* in vitro and in planta. The use of natural bioactive compounds with low phytotoxicity could be a more natural alternative against important plant pathogens such as *B. cinerea*. The ability of both native geraniol and geraniol—loaded Nano Emulsions (GNEs) to successfully control *B. cinerea* on cucumber plants was demonstrated. More specifically, the GNEs that were synthesized proved to be effective at a lower concentration compared to native geraniol, showing the importance of a controlled release of this compound. This result is furtherly supported by the conducted pharmacokinetic study that underlined that the use of SDS as an emulsifier provides long-term stability of geraniol and could be studied as a geraniol-delivery system. The metabolic profiling and the MDA levels combined with the study of the disease index demonstrated a possible role of geraniol as a signal molecule that can trigger the induction of priming effects. The LC-QTOF-MS analysis of the cucumber plants demonstrated that a total of 27 target (8) and suspect (19) bioactive compounds weredetermined. Cucumber plants treated with geraniol showed an increase in the concentration levels of myricetin, chlorogenic acid, and kaempferol rhamnoside as compared to control plants and the presence of *B. cinerea* caused an increase in sinapic acid and genistein. Further research is required to identify the potentially toxic effects of nanopesticides on the environment and on consumers’ health, as well as the molecular mechanisms of the priming effect induced by geraniol.

## Figures and Tables

**Figure 1 plants-11-02513-f001:**
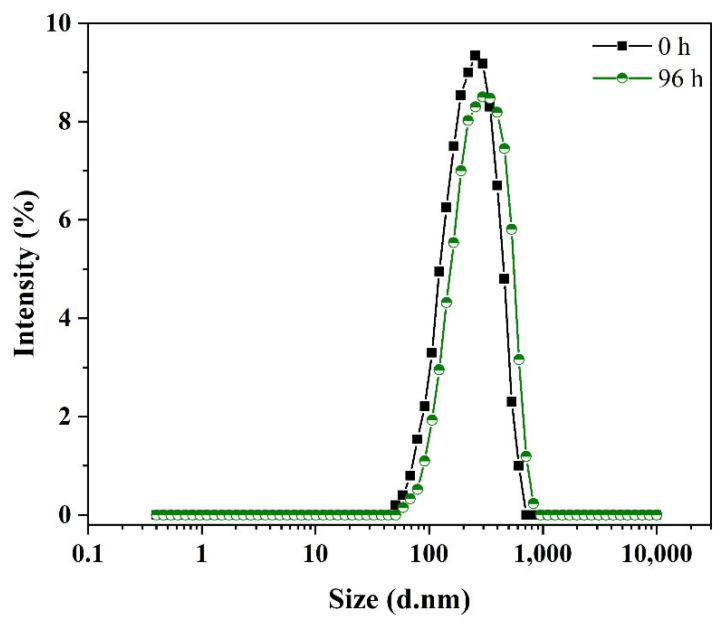
Size distribution of the geraniol-loaded Nano Emulsions (GNEs) in an aqueous solution (pH = 7.2) evaluated by Dynamic Light Scattering (DLS) technique at 0 h and 96 h (black and green lines, respectively) after preparation.

**Figure 2 plants-11-02513-f002:**
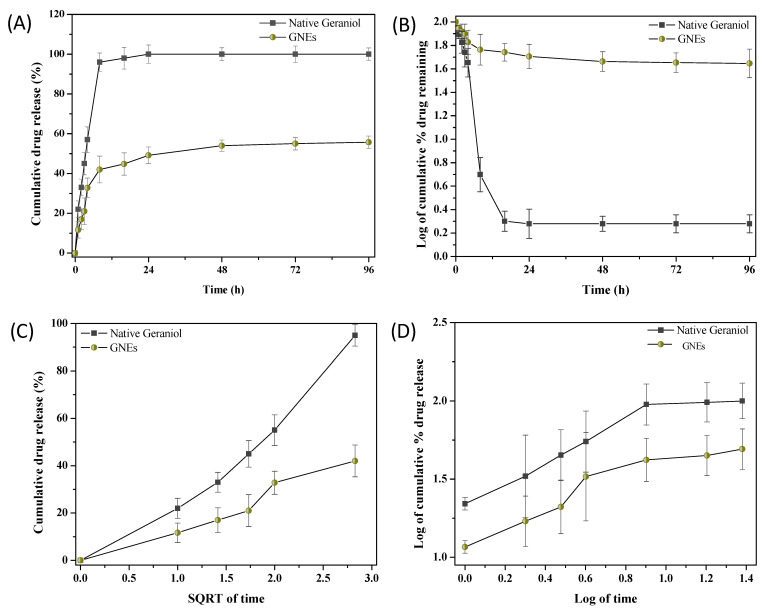
Comparative study of native geraniol and geraniol-loaded Nano Emulsions (GNEs) release in a 96 h period using the zero-order (**A**), first-order (**B**), Higuchi (**C**), and Korsmeyer–Peppas (**D**) models. Values are expressed as a mean of three replicates. Error bars represent standard deviation.

**Figure 3 plants-11-02513-f003:**
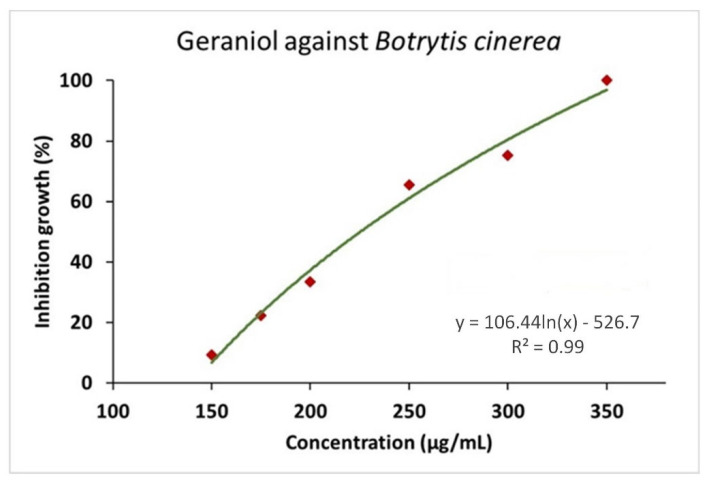
Dose–response growth inhibition curves of *B. cinerea* exposed to seven (7) different concentrations of geraniol (150, 175, 200, 225, 250, 300, and 350 μg/mL). Each point represents the mean of ten replicates per geraniol concentration (2 experiments, 5 replications), EC_50_ = 235 μg/mL.

**Figure 4 plants-11-02513-f004:**
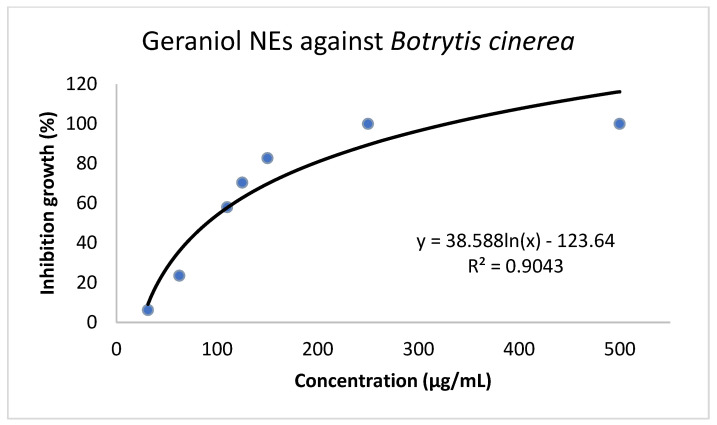
Dose–response growth inhibition curves of *B. cinerea* exposed to seven (7) different concentrations of geraniol-loaded Nano Emulsions (GNEs) (25, 75, 100, 125, 150, 250, and 500 μg/mL). Each point represents the mean of ten replicates per GNEs concentration (2 experiments, 5 replications), EC_50_ = 105 μg/mL.

**Figure 5 plants-11-02513-f005:**
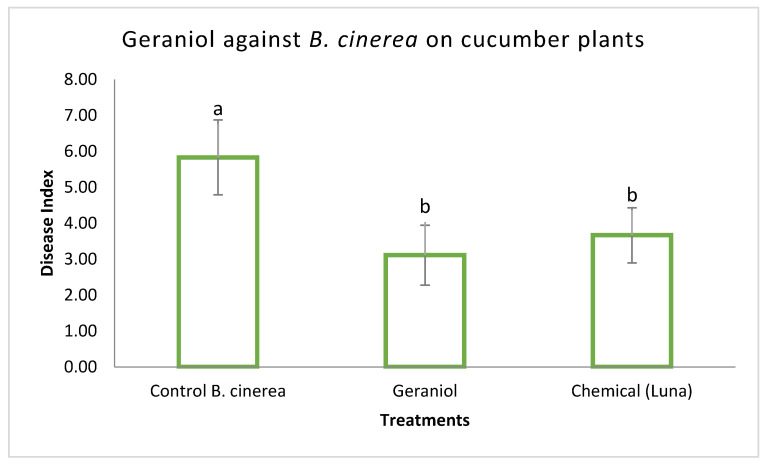
Effect of geraniol (sprayed at 470 μg/mL) and commercial fungicide (sprayed at maximal concentration) on severity of disease caused by *B. cinerea* on cucumber in pots under controlled conditions. *B. cinerea* control plants were inoculated only with the pathogen. Disease was assessed 6 days after inoculation. The experiment was repeated three times. Different letters indicate significant differences according to Tukey’s test at *p* ≤ 0.05. Error bars represent standard deviation.

**Figure 6 plants-11-02513-f006:**
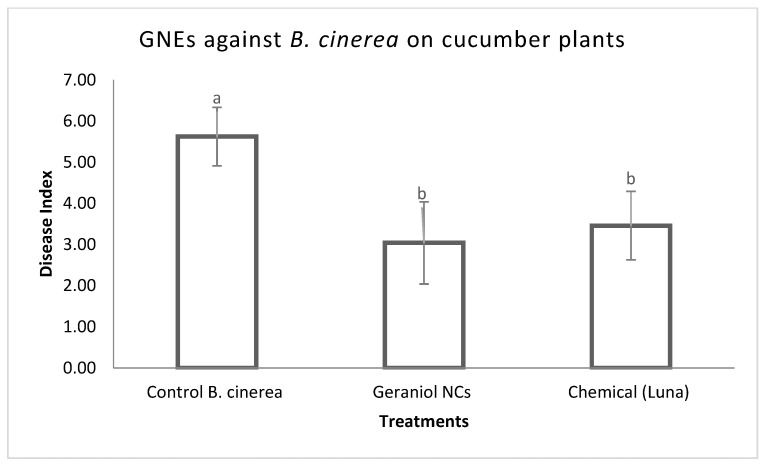
Effect of geraniol-loaded nano emulsions (GNEs) (sprayed at 210 μg/mL) and commercial fungicide (sprayed at maximal concentration) on severity of disease caused by *B. cinerea* on cucumber, in pots under controlled conditions. *B. cinerea* control plants were inoculated only with the pathogen. Disease was assessed 6 days after inoculation. The experiment was repeated three times. Different letters indicate significant differences according to Tukey’s test at *p* ≤ 0.05. Error bars represent standard deviation.

**Figure 7 plants-11-02513-f007:**
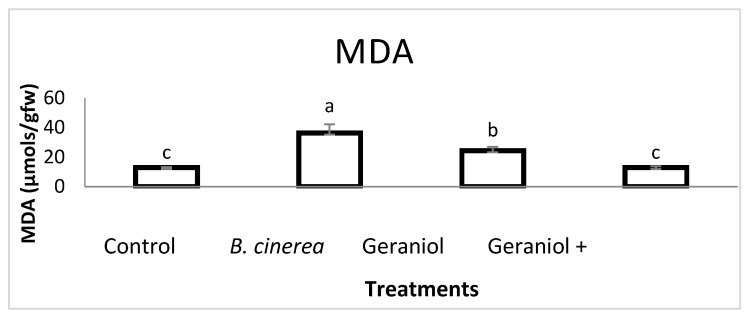
Effect of geraniol, the plant pathogenic fungus *B. cinerea* and their combination (geraniol + *B. cinerea*) on MDA content. Different letters indicate significant differences according to Tukey’s test at *p* ≤ 0.05. Error bars represent standard deviation.

**Figure 8 plants-11-02513-f008:**
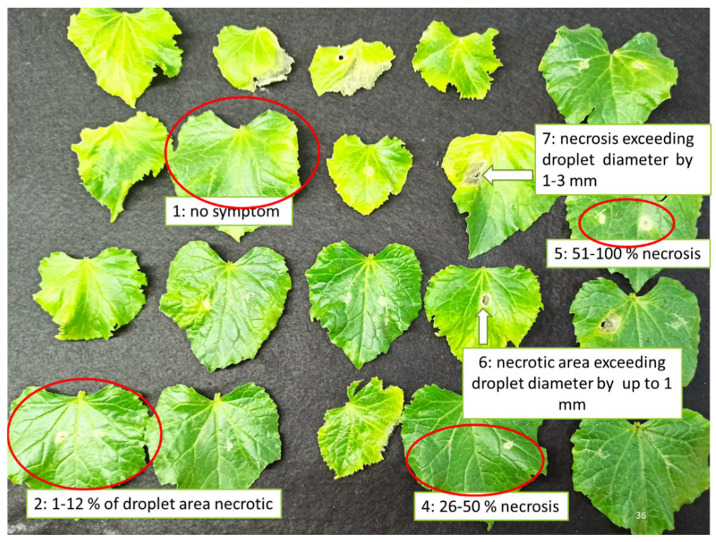
Indicative symptoms demonstrating the extent of disease caused by *B. cinerea* on cucumber leaves according to disease index (DI): 1: no symptom; 2: 1–12% of droplet area necrotic; 3: 13–25% necrosis; 4: 26–50% necrosis; 5: 51–100% necrosis; 6: necrotic area exceeding droplet diameter by up to 1 mm; 7: necrosis exceeding droplet diameter by 1–3 mm; 8: necrosis exceeding droplet diameter by more than 3 mm [68].

**Table 1 plants-11-02513-t001:** EC_50_ values based on dose–response curves of geraniol and geraniol-loaded Nano Emulsions GNEs against *Botrytis cinerea* 96 h post inoculation (hpi).

Treatments	EC_50_ Values (μg/mL)
Geraniol	235
GNEs	105

**Table 2 plants-11-02513-t002:** Target screening results.

Compound	Molecular Formula	[M-H]-	*RT (min)
Caffeic acid	C_9_H_8_O_4_	139.075	4.96
p-coumaric acid	C_9_H_8_O_3_	163.040	5.85
Ferulic acid	C_10_H_10_O_4_	193.051	6.13
Sinapic acid	C_11_H_12_O_5_	223.061	6.15
Myricitrin	C_21_H_20_O_12_	463.088	6.59
Myricetin	C_15_H_10_O_8_	317.030	7.03
Luteolin	C_15_H_10_O_6_	285.040	8.25
Apigenin	C_15_H_10_O_5_	269.046	8.96

*RT: retention time.

**Table 3 plants-11-02513-t003:** Suspect screening results and tentative identification criteria.

Compound	Molecular Formula	[M-H]-	*RT (min)	MS/MS Spectra Comparison
Coumaric acid glucoside	C_15_H_18_O_8_	325.093	4.83	FOODB record: FDB019119
Quercetin rhamnosylrutinoside	C_39_H_50_O_25_	755.204	5.01	[27]
Astragalin	C_21_H_20_O_11_	447.093	5.93	MassBank record: MSBNK-RIKEN_ReSpect-PS042211
Coreopsin	C_21_H_22_O_10_	433.114	6.04	[28]
Salicylic acid	C_7_H_6_O_3_	137.024	6.15	MassBank record: MSBNK-Keio_Univ-KO000602
Kaempferol rhamnosyl glucoside	C_33_H_40_O_20_	593.151	6.26	[27]
Tiliroside	C_30_H_26_O_13_	593.130	6.32	MassBank record: MSBNK-RIKEN-PR100968
Narcissin	C_28_H_32_O_16_	623.162	6.36	[29]
Kaempferol rhamnoside	C_21_H_20_O_10_	431.098	6.44	[27]
Phloridizin	C_21_H_24_O_10_	435.130	6.58	[30]
Hyperoside/ Isoquercitrin	C_21_H_20_O_12_	463.088	6.62	MassBank record: MSBNK-Fiocruz-FIO00168
Naringenin glucoside	C_21_H_22_O_10_	433.114	6.73	MassBank record: MSBNK-RIKEN-PR306421
Isorhamnetin glucoside	C_22_H_22_O_12_	477.104	7.25	MassBank record: [MSBNK-RIKEN-PR040095]
Chlorogenic acid	C_16_H_18_O_9_	353.088	7.85	[28]
Coniferaldehyde	C_10_H_10_O_3_	177.056	7.99	MassBank record: MSBNK-RIKEN_ReSpect-PT200060
Sinapaldehyde	C_11_H_12_O_4_	207.066	8.09	MassBank record: MSBNK-RIKEN-PR309000
Naringenin	C_15_H_12_O_5_	271.161	8.22	MassBank record: MSBNK-IPB_Halle-PN000004
Genistein	C_15_H_10_O_5_	269.046	8.98	MassBank record: MSBNK-MSSJ-MSJ00976
Diosmetin	C_16_H_12_O_6_	299.056	9.06	MassBank record: MSBNK-BS-BS003183

*RT: retention time.

**Table 4 plants-11-02513-t004:** Compounds that showed a significant increase following treatments with geraniol, *B. cinerea,* and their combination (geraniol + *B. cinerea*), as compared to control plants. Each number represents the mean of 5 replicates in mg/kg.

Compounds	Cucumber Control	Geraniol Control	*B. cinerea* control	*B. cinerea* + geraniol
Sinapic acid	* 0.5876 ^a^ ± 0.08	0.6288 ^ab^ ± 0.76	0.9812 ^b^ ± 0.09	0.9584 ^b^ ± 0.18
Myricetin	0.2156 ^a^ ± 0.001	0.2178 ^b^ ± 0.001	0.2155 ^a^ ± 0.001	0.2167 ^ab^ ± 0.001
Chlorogenic acid	100.59 ^a^ ± 2.74	152.54 ^c^ ± 6.55	120.31 ^ab^ ± 2.22	147.96 ^bc^ ± 18.47
Genistein	0.7252 ^a^ ± 0.001	0.7199 ^a^ ± 0.002	0.7332 ^b^ ± 0.002	0.7359 ^b^ ± 0.003
Kaempferol rhamnoside	0.9080 ^a^ ± 0.002	1.1860 ^b^ ± 0.02	0.9480 ^a^ ± 0.004	1.7620 ^c^ ± 0.14

* Different letters in each line indicate significant differences according to Duncan’s test at *p* ≤ 0.05 ± Standard Error.

**Table 5 plants-11-02513-t005:** Regression equation and correlation coefficients of target compounds.

Compound	Type	Calibration Equation	R^2^
Caffeic acid	Linear	y = 4E + 06x + 49264	0.998
p-coumaric acid	Linear	y = 2E + 06x + 187478	0.995
Ferulic acid	Linear	y = 1E + 06x + 45615	0.996
Sinapic acid	Linear	y = 151632x − 142.29	0.999
Myricitrin	Linear	y = 30610x − 1114.8	0.999
Myricetin	Linear	y = 8E + 06x − 344334	0.997
Luteolin	Linear	y = 1E + 07x + 721454	0.996
Apigenin	Linear	y = 7E + 07x + 3E + 06	0.991

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
