# Peer review of "Impact of Geraniol and Geraniol Nanoemulsions on Botrytis cinerea and Effect of Geraniol on Cucumber Plants’ Metabolic Profile Analyzed by LC-QTOF-MS"

_plants, 2022, doi:10.3390/plants11192513_

Round 1
Reviewer 1 Report
The manuscript under consideration by Kamou et al is a well-designed study of importane problem of agrochemistry. Despite the fact that cucmber could be considered as a trivial object gray mold is still a big challenge for agrotechnology.
The manuscript is in scope of Plants and can be published after minor changes.
1) I would recomend to add a graphical abstract with the scheme of manuscript workflow: a lot of data are presented and for better representation any kind of navigation scheme is recommended
2) It is very important to enrich all Figures and Tables names with all details and conditions
3) Is it possible to enlarge the developed formulation to fight with other agricultural issues? Probably for other plants?
4) Will the garaniol influence on the organoleptic properties of cucumbers? Will the veggies produced with this nanoemulsion be named "organic"?
Otherwise I would recomend to publish this manuscript after minor changes
Author Response
Reviewer 1:
The manuscript under consideration by Kamou et al is a well-designed study of importane problem of agrochemistry. Despite the fact that cucmber could be considered as a trivial object gray mold is still a big challenge for agrotechnology.
The manuscript is in scope of Plants and can be published after minor changes.
1) I would recomend to add a graphical abstract with the scheme of manuscript workflow: a lot of data are presented and for better representation any kind of navigation scheme is recommended
A graphical abstract was added
2) It is very important to enrich all Figures and Tables names with all details and conditions
All figures and tables were corrected according to this comment
3) Is it possible to enlarge the developed formulation to fight with other agricultural issues? Probably for other plants?
There is currently another project under progress that will be published soon covering tomato, two other plant pathogens and nematodes, thus presenting the ability of these (and other similar) formulations to address other important agricultural issues.
4) Will the garaniol influence on the organoleptic properties of cucumbers? Will the veggies produced with this nanoemulsion be named "organic"?
The organoleptic properties of cucumbers after treatment with geraniol were studied (data not published). The diameter (mm), weight (gr), length (cm), titratable acidity and Brix values (% soluble solids content) were measured/studied in order to define the effect of geraniol on cucumbers. The overall results showed that these properties are slightly (not statistically significant) affected positively. Regarding the organic labelling of the produce: the use of plant protection products in organic farming undergoes a legislative procedure and such product could be a possible candidate for this kind of certification.
Otherwise I would recomend to publish this manuscript after minor changes
Reviewer 2 Report
The work in this document is of an acceptable standard for publication in this journal because it is a topic that is not popular and pharmacokinetic was study
-Line 98 : (SDS is an anionic surfactant that penetrates the oil-droplet and forms stable core-shell nanoparticles when used at high concentrations)
The justifications must be made in the discussion not in the results section
-Line 104-105 : due to Ostwald 104 ripening since geraniol has some degree of water solubility because of the hydroxyl group in their phenolic structure
The justifications must be made in the discussion not in the results section
-Line 337 : Botrytis cinerea strain B05 : Which?
Line 356 : (R – r)/R × 100 (%) This is not an equation, use the normal form of equation in formula
-Line 359 : In planta antifungal activity : Give information about the method used reference
-Line 392 : LC-QTOF-MS analysis : Since geraniol is volatile terpene compound, justify why you didn't use GC and you had used LC?

Author Response
Reviewer 2:
Comments and Suggestions for Authors
The work in this document is of an acceptable standard for publication in this journal because it is a topic that is not popular and pharmacokinetic was study
-Line 98 : (SDS is an anionic surfactant that penetrates the oil-droplet and forms stable core-shell nanoparticles when used at high concentrations)
The justifications must be made in the discussion not in the results section
This sentence was deleted from the results section and added to the discussion section. Reference numbers were changed accordingly (lines 242 – 247).
-Line 104-105 : due to Ostwald 104 ripening since geraniol has some degree of water solubility because of the hydroxyl group in their phenolic structure
The justifications must be made in the discussion not in the results section
This sentence was deleted from the results section and added to the discussion section. Reference numbers were changed accordingly (lines 242 – 247).
-Line 337 : Botrytis cinerea strain B05 : Which?
This strain was obtained from the collection of the Plant Pathology Laboratory of Aristotle University of Thessaloniki and its number was added (B 05.10) (Samaras et al., 2021, Insights into the multitrophic interactions between the biocontrol agent Bacillus subtilis MBI 600, the pathogen Botrytis cinerea and their plant host, Microbiological Research, 248, https://doi.org/10.1016/j.micres.2021.126752.)
Line 356 : (R – r)/R × 100 (%) This is not an equation, use the normal form of equation in formula
Equation was added using the normal form (line 355)
-Line 359 : In planta antifungal activity : Give information about the method used reference
The reference was added (line 364). The modifications of this method are stated in this section (lines 364 – 366).
-Line 392 : LC-QTOF-MS analysis : Since geraniol is volatile terpene compound, justify why you didn't use GC and you had used LC?
The aim of the analysis was focused on the metabolites produced by the cucumber plant after application of geraniol and not on determining the geraniol compound itself. This is why we chose LC-QTOF-MS for this study
Reviewer 3 Report
Comments
Kamou and co-workers' research findings on the impact of geraniol and geraniol nanoemulsions on Botrytis cinerea and the effect of geraniol on cucumber plants’ metabolic profile using LC-QTOF-MS. The authors provided some good results about the inhibition of B. cinerea under greenhouse conditions. The authors should improve the discussion section of this manuscript. In addition, I have the following comments that the authors may want to take into account to improve the quality of their manuscript:
1. Why did you specifically choose geraniol in this study? So add more details about the importance of geraniol in the introduction section.
2. In Table 2, for tentative identification of compounds, add citations for respective compounds.
3. In Table 3, the authors mentioned 19 compounds based on suspect screening, but only 5 components were presented in Table 4. On what basis, were these five compounds selected for quantification? This part also should be explained well in the discussion section.
4. Add footnotes for Figure 8 and Tables 2 - 4.
5. Cite and discuss the following article in this manuscript.
Gago CML, Artiga-Artigas M, Antunes MDC, Faleiro ML, Miguel MG, Martín-Belloso O. Effectiveness of nanoemulsions of clove and lemongrass essential oils and their major components against Escherichia coli and Botrytis cinerea. J Food Sci Technol. 2019; 56(5): 2721-2736.
6. The discussion section should be improved by adding details on the antifungal potential of geraniol.
7. Use abbreviation for Botrytis cinerea (B. cinerea) – lines 230, 337, and 477.
8. What is the highest recommended dose of the chemical fungicide, Luna?
9. What about the phytotoxic concentration of geraniol?
10. Figures 5 to 7, clearly present these figures with X and Y axes lines.
11. Correct Figure 7 - Geraniol +
12. Italicize “Botrytis cinerea” - Reference No. 59.
Author Response
Reviewer 3:
Kamou and co-workers' research findings on the impact of geraniol and geraniol nanoemulsions on Botrytis cinerea and the effect of geraniol on cucumber plants’ metabolic profile using LC-QTOF-MS. The authors provided some good results about the inhibition of B. cinerea under greenhouse conditions. The authors should improve the discussion section of this manuscript. In addition, I have the following comments that the authors may want to take into account to improve the quality of their manuscript:
- Why did you specifically choose geraniol in this study? So add more details about the importance of geraniol in the introduction section.
Details underlining the importance of geraniol were added in the introduction (lines 60 – 62).
- In Table 2, for tentative identification of compounds, add citations for respective compounds.
The reviewer must mean table 3. Citations were added according to the comment (lines 183 – 185) and within Table 3.
- In Table 3, the authors mentioned 19 compounds based on suspect screening, but only 5 components were presented in Table 4. On what basis, were these five compounds selected for quantification? This part also should be explained well in the discussion section.
Table 4 demonstrates compounds that showed a significant increase following treatments with geraniol, B. cinerea and their combination (geraniol + B. cinerea), as compared to control plants, in order to underline the biological importance of these treatments. Whereas Table 2 and 3 demonstrate all compounds that were successfully identified. This part is fully explained in lines 262-281 of the Discussion section.
- Add footnotes for Figure 8 and Tables 2 - 4.
Captions and footnotes were changed according to comment
- Cite and discuss the following article in this manuscript.
Gago CML, Artiga-Artigas M, Antunes MDC, Faleiro ML, Miguel MG, Martín-Belloso O. Effectiveness of nanoemulsions of clove and lemongrass essential oils and their major components against Escherichia coli and Botrytis cinerea. J Food Sci Technol. 2019; 56(5): 2721-2736.
The following article was cited and discussed (lines 245 – 247).
- The discussion section should be improved by adding details on the antifungal potential of geraniol.
The Discussion and Introduction were changed according to this comment, references were also added (lines 60 – 62 and 234 – 237).
- Use abbreviation for Botrytis cinerea(B. cinerea) – lines 230, 337, and 477.
Abbreviations were corrected
- What is the highest recommended dose of the chemical fungicide, Luna?
Luna® Sensation SC Fungicide (Bayer Crop Science, fluopyram 250 g/L and trifloxystrobin 250 g/L). Highest recommended dose for vegetables: 30 ml/str (max) with 75 L injection liquid/acre, according to the label.
- What about the phytotoxic concentration of geraniol?
A study focusing on the organoleptic properties of cucumber after treatment with geraniol, at the same concentrations as described in this MS, was performed (data not published). The diameter (mm), weight (gr), length (cm), titratable acidity and Brix values (% soluble solids content) were measured/studied in order to define the effect of these substances on cucumbers. The overall results showed that these properties are slightly (not statistically significant) affected positively. These results supported the fact that with the concentrations we used not only we did not have any phytotoxicity, but the plants were improved. We are aware of the potential phytotoxicity attributed to EOs and their constituents and we also consider that cucumber is quite sensible, that is why these results were so positive.
- Figures 5 to 7, clearly present these figures with X and Y axes lines.
The axes were added on the figures
- Correct Figure 7 - Geraniol +
This figure was corrected
- Italicize “Botrytis cinerea” - Reference No. 59.
The correction was applied
Round 2
Reviewer 3 Report
I have read the reply of the authors and the revised manuscript. I can say that the authors answered most of the inquiries about their work. Further, the manuscript has been modified according to the reviewers' comments and suggestions. Hence, I recommend this manuscript for publication in Plants Journal.